# Bioactivation Treatment with Mixed Acid and Heat on Titanium Implants Fabricated by Selective Laser Melting Enhances Preosteoblast Cell Differentiation

**DOI:** 10.3390/nano11040987

**Published:** 2021-04-12

**Authors:** Phuc Thi Minh Le, Seine A. Shintani, Hiroaki Takadama, Morihiro Ito, Tatsuya Kakutani, Hisashi Kitagaki, Shuntaro Terauchi, Takaaki Ueno, Hiroyuki Nakano, Yoichiro Nakajima, Kazuya Inoue, Tomiharu Matsushita, Seiji Yamaguchi

**Affiliations:** 1Department of Biomedical Sciences, College of Life and Health Sciences, Chubu University, 1200 Matsumoto, Kasugai, Aichi 487-8501, Japan; shintani@isc.chubu.ac.jp (S.A.S.); takadama@isc.chubu.ac.jp (H.T.); m-ito@isc.chubu.ac.jp (M.I.); matsushi@isc.chubu.ac.jp (T.M.); 2Osaka Yakin Kogyo Co., Ltd., Zuiko 4-4-28, Higashi Yodogawa-ku, Osaka City, Osaka 533-0005, Japan; kakutani@e.osakayakin.co.jp (T.K.); kitagaki@e.osakayakin.co.jp (H.K.); terauchi@e.osakayakin.co.jp (S.T.); 3Department of Dentistry and Oral Surgery, Division of Medicine for Function and Morphology of Sensor Organ, Faculty of Medicine, Dentistry and Oral Surgery, Osaka Medical College, 2-7 Daigaku-machi, Takatsuki City, Osaka 569-8686, Japan; ueno@osaka-med.ac.jp (T.U.); ora099@osaka-med.ac.jp (H.N.); n4160@osaka-med.ac.jp (Y.N.); ora092@osaka-med.ac.jp (K.I.)

**Keywords:** selective laser melting, acid treatment, titanium, apatite formation, cell culture

## Abstract

Selective laser melting (SLM) is a promising technology capable of producing individual characteristics with a high degree of surface roughness for implants. These surfaces can be modified so as to increase their osseointegration, bone generation and biocompatibility, features which are critical to their clinical success. In this study, we evaluated the effects on preosteoblast proliferation and differentiation of titanium metal (Ti) with a high degree of roughness (Ra = 5.4266 ± 1.282 µm) prepared by SLM (SLM-Ti) that was also subjected to surface bioactive treatment by mixed acid and heat (MAH). The results showed that the MAH treatment further increased the surface roughness, wettability and apatite formation capacity of SLM-Ti, features which are useful for cell attachment and bone bonding. Quantitative measurement of osteogenic-related gene expression by RT-PCR indicated that the MC3T3-E1 cells on the SLM-Ti MAH surface presented a stronger tendency towards osteogenic differentiation at the genetic level through significantly increased expression of Alp, Ocn, Runx2 and Opn. We conclude that bio-activated SLM-Ti enhanced preosteoblast differentiation. These findings suggest that the mixed acid and heat treatment on SLM-Ti is promising method for preparing the next generation of orthopedic and dental implants because of its apatite formation and cell differentiation capability.

## 1. Introduction

Titanium metal (Ti) and its alloys are widely used in the orthopedic and dental surgery fields because they possess a high corrosion resistance, good biocompatibility, and low elastic moduli closer to that of natural bone than other metals such as stainless steel and cobalt chromium alloy [1,2,3], but they do not bond directly to living bone and are difficult to shape [4,5]. Controlling the surface and morphological characteristics for each implant so that it may be fit with the excision needs of surgical procedures has been challenging. Selective laser melting (SLM) on Ti is a promising technology for solving these difficulties. In the SLM technique, small Ti particles are melted and fused layer by layer using a high-intensity laser, and computer-aided design was applied to make three-dimensional structures of the materials [6,7]. Therefore, SLM used to make materials with a high surface roughness and personalized to the needs of the bone defect(s) of each patient. At present, SLM-Ti materials are commonly used in various research efforts in the medical and dental fields, to assess cellular behaviors in vitro [8,9] or osseointegration in vivo [10,11]. Implants fabricated by SLM are in clinical trials for patients with facial bone defects [12,13].

Because Ti and its alloys themselves do not bond directly to the living bone, various surface treatments such as hydroxyapatite coating [14], peptide immobilization [15], and bio-activation by simple chemical and thermal procedures [16,17,18] have been developed to promote this bonding. Of these treatments, chemical and thermal procedures are the most convenient because they do not require special equipment, can treat in a uniform manner and may be applied to a complex surface, inducing bone in the deep valleys of the material [19,20]. Tsukanaka et al. and Pattanayak et al. reported dense and porous SLM-Ti materials treated with dilute HCl and heat following NaOH treatment promoted osteoblastic differentiation in vitro and increased bone formation in vivo [21,22]. Kokubo et al. and Kawai et al. proved a bioactive treatment with Ti by a mixture of H_2_SO_4_ and HCl and subsequent heat increased its osteogenic capacity to exhibit osteoconductivity and osteoinductivity [23,24,25]. A certain kind of orthopedic implant treated with a bioactive treatment such as NaOH and heat treatment is already used in the clinical and has obtained good osteoconduction without any adverse effects [26,27]. Sandblasting and subsequent acid-etching have produced roughened surfaces on Ti on a scale of several dozen micrometers as well as a few micrometers and are in clinical use in the dental field [28,29]. Although it increases both bone-implant contact and cell adhesion, it does not always provide stable fixation on the Ti implant for an extended period because of the poor mineralization capacity of the treated metal [30,31].

In our previous in vivo studies, the Ti mesh that was prepared by SLM treated with mixed acid and heat promoted bone formation in rat calvarial bone defects [32,33]. However, the influence of the highly roughened surface of the SLM-Ti materials in combination with the mixed acid and heat treatment effect on cellular was not elucidated in vitro. Theoretically, the mixed acid and heat treatment can further increase cell adhesion, proliferation, and differentiation due to the increased surface roughness, wettability, and apatite formation of Ti.

In this study, we evaluated the effects of a high degree of surface roughness of the SLM-Ti materials with or without the mixed acid treatment of H_2_SO_4_ and HCl and subsequent heat treatment on cellular behaviors such as cell morphology, cell viability, and osteogenic differentiation. The impact of the surface treatment on the osteogenic gene was also evaluated by examining gene expression level of osteogenic differentiation-related genes such as alkaline phosphatase (Alp), Runx2, Osteocalcin (Ocn), and osteopontin (Opn) as well as β-catenin, integrin β1, cyclin D1. The results were compared with those obtained with polished cp-Ti with a smooth surface. This study provides insight into the importance of the role of the high surface roughness and bioactive treatment on cell behavior.

## 2. Materials and Methods

### 2.1. Sample Preparation

Four types of samples were prepared, as follows. SLM-Ti discs 18 mm in diameter and of 1 mm thickness (Osaka Yakin Kogyo Co., Ltd., Osaka, Japan) were modified using an EOSINT M270 SLM machine (Electro Optical Systems, Krailing, Germany) employing commercially pure titanium metal powder (>99.5% pure) with particle diameters of less than 45 µm (Osaka Titanium Technologies, Amagasaki, Hyogo, Japan). The discs were subjected to ultrasonic cleaning using acetone, 2-propanol, and ultrapure water for 30 min at each step (SLM-Ti U). The cleaned samples were immersed in a mixture of 66.3% H_2_SO_4_ and 10.3% HCl solutions at 70 °C for 1 h (SLM-Ti MA). They were then heated at 600 °C for 1 h in an electric furnace and allowed to be naturally cooled (SLM-Ti MAH). Cp-Ti discs that were 18 mm diameter and 1 mm thickness were cut from commercially pure (99.5%) titanium plates (Nilaco, Tokyo, Japan) and polished with grade 400 diamond plates.

### 2.2. Surface Characteristics

The surface topography of the prepared materials was observed under field emission scanning electron microscopy (FE-SEM, S-4300, Hitachi, Tokyo, Japan) at an accelerator voltage of 15 kV.

The roughness parameters on the sample surface were measured using a Mitutoyo Surftest SV-2000 instrument (Mitutoyo America Corporation, Aurora, IL, USA). Five random positions on the surfaces of SLM-Ti and cp-Ti were measured with a measurement length of 8000 µm. The surface roughness values are shown as the mean ± standard deviation of Ra, the average peak to valley distance; and Rz, the distance from the highest peak to the lowest valley.

The crystalline phase of the samples analyzed by thin-film X-ray diffraction (TF-XRD, RNT-2500, Rigaku Co., Tokyo, Japan). The measurement was conducted at a power of 50 kV and 200 mA, CuK-alpha was used as an X-ray source, and the incident beam angle was set at 1° against the sample surface.

The distribution of various elements near the sample surface was examined by Radio frequency (RF) glow discharge optical emission spectroscopy (GD-OES, GD-Profiler 2, Horiba Co., Kyoto, Japan) under Ar sputtering at an Ar pressure of 600 Pa.

The water contact angle of the samples was measured to determine their hydrophobic characteristics. A 4 μL droplet of ultrapure water was dropped onto the surface using a pipette. A photograph of the droplet on the Ti surfaces was taken and the contact angle was calculated from this image.

### 2.3. Apatite Formation

The SLM-Ti with or without mixed acid and heat along with the control cp-Ti samples were submerged in pre-warmed simulated body fluid (SBF) with ion concentrations nearly equal to those of human blood plasma [34] for 1 day at 36.5 °C. The apatite formation on the surface was then observed by FE-SEM.

### 2.4. Cell Culture

MC3T3-E1 cells (CRL-2594, ATCC, Manassas, VA, USA) were cultured at 37 °C, 5% CO_2_ in Minimum Essential Medium Eagle-Alpha Modification (Alpha MEM, Gibco, Thermo Fisher Scientific, Waltham, MA, USA) supplemented with 10% (*v/v*) fetal bovine serum, 1% (*v/v*) penicillin and streptomycin. Cells were subcultured after 2–3 days when the confluency had reached 70–80%. The cells were seeded on the four types of materials described in Section 2.1 to investigate the influence of surface roughness and bioactive treatment on cell behavior. All of the samples were sterilized by ethylene oxide gas before being subjected to the cell experiments.

### 2.5. Cell Viability and Cell Morphology

The cells were seeded on SLM-Ti and cp-Ti discs on a 12-well plate at a density 1.0 × 10^5^ cells/1.0 mL of medium/well and then cultured in MEM at 37 °C, 5% CO_2_. After 1 and 3 days, cell viability was evaluated using the cell count reagent SF (Nacalai Tesque, Kyoto, Japan) according to the manufacturer’s instructions. This is a colorimetric assay using a highly water-soluble tetrazolium salt which is reduced to formazan dye in living cells. After incubation for 2 h, the optical density was measured at 450 nm using a microplate reader (iMarkTM, Bio-Rad, Hercules, CA, USA).

For determination of the cell morphology, 3 × 10^4^ cells/1.0 mL of the medium/well were cultivated on Ti discs at 37 °C, 5% CO_2_. After seeding for 0.5 h, 1 h, and 24 h, the non-adherent cells were removed by rinsing with phosphate-buffered saline (PBS) solution. Cells were fixed with 2.5% glutaraldehyde at 4 °C overnight, dehydrated in gradually increased alcohol concentrations from 50% to 100%, and dried under a vacuum condition. The samples were then subjected to carbon coating followed by SEM observation to evaluate cell morphology.

### 2.6. Alkaline Phosphatase Activity

Alkaline phosphatase (Alp) activity assay to assess osteogenesis and cellular differentiation was performed as follows. The cells were seeded on Ti discs on 12-well plates at a density of 1.0 × 10^5^ cells/1.0 mL of the medium/well at 37 °C, 5% CO_2_ until the cells became fully confluent. After that a new culture medium was exchanged for the old one and the cells continued to be incubated for 3, 10, and 14 days. The cells were washed twice with PBS and lysed with 0.05% Triton X-100 through two standard freeze-thaw cycles. The lysis solution was then transferred to Eppendorf tubes and centrifuged at 15,000 rpm for 15 min at 4 °C. The supernatant was used for assessing the Alp activity using LabAssay^TM^ ALP, a colorimetric assay using an Alp reagent containing *p*-nitrophenyl phosphate (Wako, Osaka, Japan). The absorbance of the p-nitrophenol that formed was measured at a wavelength of 405 nm. Alp activity was normalized against the total protein concentration, which was determined with a BCA kit (Thermo Fisher Scientific), using bovine serum albumin as the standard.

### 2.7. Extracellular Matrix Mineralization

Extracellular calcium deposition is an indication of successful primary osteoblast differentiation, and calcium deposits can be specifically stained bright orange-red by Alizarin Red staining. 1.0 × 10^5^ cells were seeded on the Ti discs in a 12-well plate until the cells became fully confluent. Then the medium was changed to an osteogenic medium containing 10 µM β-glycerophosphate and 50 µg/mL ascorbic acid. After 3 weeks of culture, the cells were washed twice with PBS, fixed in 4% paraformaldehyde for 20 min at room temperature, and washed with distilled water five times. The cells were then stained with 40 mM Alizarin Red in distilled water for 20 min in a dark at room temperature. Afterward, the cells were washed with distilled water until no color appeared, and then the samples were dried. For quantitation of mineralized matrix, stained calcium nodules were dissolved in 10% cetylpyridinium chloride solution and the absorbance values were measured at 570 nm. The values were expressed relative to that of cp-Ti.

### 2.8. Real-Time PCR for Assessing Gene Expression

1.0 × 10^5^ cells were cultured on Ti discs in 12-well plates at 37 °C and 5% CO_2_, then cultured for 7 and 14 days. Total RNA was extracted using the aurum total RNA kit (BioRad, Hercules, CA, USA) following the manufacturer’s instruction, and reverse-transcribed with the iScript advanced cDNA synthesis kit for RT-qPCR (BioRad,). RT-qPCR was carried out on the Biorad CFX96 Touch real-time PCR with iTaq universal SYBR Green Supermix. The primer sequences are listed in Table 1. The gene expression level of β-catenin, integrin β1, cyclin D1, and osteogenic differentiation-related genes such as alkaline phosphatase (Alp), Runx2, osteocalcin (Ocn), and osteopontin (Opn) was normalized to that of GADPH.

### 2.9. Statistical Analysis

All data were assessed for statistical significance using the statistical software program Microcal Origin 8.5.1 (Microcal Software, Inc., Northampton, MA, USA). The differences between the mean values of the different groups were determined using one-way ANOVA followed by a Tukey post-hoc test. All values are presented as the mean ± standard derivation (sd) with at least three independent replicates (n ≥ 3), *p* < 0.05 was considered to be significant.

## 3. Results

### 3.1. The Surface Characteristics

Representative SEM images of all of the materials used are shown in Figure 1. The surface of the cp-Ti is seen to be flat in the magnified image while the surface of the SLM-Ti disc exhibited a particular morphology due to the material-derived particles that were partially melted and remained on the metal surface even after the ultrasonic cleaning. The mixed acid treatment further increased the surface roughness by producing a micro-meter porous architecture typical of acid-etching and removal of the partially-melted, small diameter particles. This surface morphology remained even after the heat treatment. The corresponding surface roughness parameters Ra and Rz shown in Figure 2 confirmed that the SLM-Ti had a larger surface roughness value (Ra = 5.4266 ± 1.282 µm, Rz = 33.515 ± 6.923 µm) than cp-Ti (Ra = 0.466 ± 0.020 µm, Rz = 3.253 ± 0.207 µm).

In addition, treatment with MA or MAH significantly increased the surface roughness of SLM-Ti; the Ra values were 8.585 ± 1.117 µm and 8.612 ± 0.990 µm, while the Rz values were 50.556 ± 3.295 µm and 49.950 ± 3.495 µm for SLM-Ti MA or SLM-Ti MAH, respectively. There were no differences in roughness between the SLM-Ti treated with mixed acid only or mixed acid and heat.

According to XRD, SLM-Ti displayed an α-Ti (PDF 01-089-4893) similar to that of cp-Ti, as shown in Figure 3. The mixed acid treatment formed titanium hydride (PDF 01-078-2951) on SLM-Ti. This phase was transformed into rutile type titanium dioxide (PDF 00-004-0551) by the heat treatment.

It can be seen from the GD-OES profile of the SLM-Ti samples in Figure 4 that the mixed acid treatment induced some amount of hydrogen into the sample surface. It can be also seen that some amount of the S and Cl that were derived from mixed acid solution were adsorbed onto the SLM-Ti. Subsequent heat treatment induced a large amount of oxygen into the sample surface, while the S and Cl remained.

Measurement of the water contact angle revealed the hydrophobic nature of the cp-Ti metal surface at 92°. The contact angle was similar on SLM-Ti U (95°) but significantly decreased to less than 1 degree on SLM-Ti MA and SLM-MAH, indicating the super hydrophilic nature of the chemically-activated surfaces.

### 3.2. Capacity for Apatite Formation

When all four types of Ti samples were soaked in SBF, only the SLM-Ti MAH fully formed spherical precipitates on its surface, while the other three samples did not exhibit any deposition. High magnification image (Figure 5) shows that the spherical precipitates compose of scale-like crystal that is typically seen in the bone-like apatite formed in SBF [34]. These results show that the heat treatment following the acid treatment has a key role in apatite formation, which is an indicator of bone-bonding.

### 3.3. Cell Morphology

There are critical differences in cell morphology between cp-Ti and SLM-Ti. The cells on cp-Ti were round in shape, while on SLM-Ti they were elongated and bridged the corners, even after incubation periods of 0.5 h. Although the elongation on untreated SLM-Ti was not increased by prolongation of the incubation periods up to 24 h, it was further increased by the mixed acid and subsequent heat treatment, as shown in Figure 6.

### 3.4. Cell Viability

Cell viability on the Ti discs during the first 3 days of incubation was assessed and is illustrated in Figure 7. Cell viability on all of the SLM-Ti was significantly increase compared to cp-Ti at all of the tested time points. In the SLM-Ti group, there is no obvious difference between SLM-Ti U or SLM-Ti MA and SLM-Ti MAH on day 1. On day 3, cell viability was largely increased on SLM-Ti MAH as well as SLM-Ti U while the changes on SLM-Ti MA and cp-Ti were small. As a result, a significant difference was observed between SLM-Ti MAH and SLM-Ti MA.

### 3.5. Alp Activity

Alp activity of cells cultured on the four types of Ti surfaces displayed no difference after 3 and 7 days (Figure 8). In contrast, the Alp activity of cells plated on SLM-Ti MA or SLM-Ti MAH after 10 and 14 days increased significantly compared to that of the cp-Ti and SLM-Ti U samples. There was no difference in Alp activity between SLM-Ti treated with MA or MAH.

### 3.6. Extracellular Matrix Mineralization

Alizarin red staining was used to evaluate calcium-rich deposits in cell culture at the end of the experimental period (on day 21). Optical density at 570 nm was measured, as shown in Figure 9. The results indicated that matrix mineralization was highest on SLM-Ti MAH followed by SLM-Ti MA and SLM-Ti U. The value on cp-Ti with a smooth surface was significantly lower than that of the other samples.

### 3.7. Gene Expression Level

Real-time PCR was used to obtain gene expression profiling of MC3T3-E1 cells related to osteogenic differentiation, such as Alp, Runx2, Ocn and Opn on the SLM-Ti samples, and results were compared to those on cp-Ti. The results in Figure 10 and Figure 11 indicated that the expression level of these genes was significantly increased in SLM-Ti MAH compared to cp-Ti on day 7 and further increased on day 14. In contrast, most of the gene expression levels other than Runx2 remained at a low level on SLM-Ti U and even on SLM-Ti MA.

We also examined the expression of integrin β1, which is a key factor involved in cell adhesion, and of β-catenin, which is a key intracellular factor in the Wnt/β-catenin osteoblast differentiation signaling pathway, and of cyclin D1, which is closely involved in cell proliferation. SLM-Ti MAH induced significantly higher β-catenin and cyclin D1 expression on day 7. In contrast, there were no differences in the expression of integrin β1 between the four types of surfaces after 7 days. A remarkable decrease compared with cp-Ti was observed on both SLM-Ti U and SLM-Ti MAH after 14 days.

## 4. Discussion

The physical and chemical properties of the surface of an implant, such as wettability, electrical charge, chemical composition, roughness and topography exert potent effects on cell adhesion, proliferation and differentiation in vitro [35,36,37]. Modifications of the surface roughness at the micro- or nano-scale with Ra values of approximately 3–5 µm by acid etching or sandblasting have been shown to be effective in promoting osteoblast differentiation in vitro and osseointegration in vivo compared to smoother surfaces with Ra values < 1 µm [38,39,40,41]. In this study, the effects of high surface roughness of SLM-Ti (Ra = 5.4266 ± 1.282 µm) and additional bioactive treatment by mixed acid and heat on the cellular responses of MC3T3-E1 preosteoblasts was investigated. The morphology, proliferation, differentiation, mineralization and gene expression of cells were compared on three different SLM-Ti surfaces; SLM-Ti U, SLM-Ti MA and SLM-Ti MAH. Cp-Ti with a low level of surface roughness (Ra = 0.466 ± 0.020 µm) was used as a control.

A great difference in surface roughness among all of the materials used was graphically illustrated by SEM images and surface roughness measurements (Figure 1 and Figure 2), in which the SLM-Ti surface roughness was shown to be higher than that of cp-Ti. The high SLM-Ti surface roughness is due to the strongly adherent, partially-melted Ti particles. Furthermore, treatment with a mixture of H_2_SO_4_/HCl and subsequent heat further increased the surface roughness of SLM-Ti by producing a micrometer-scale roughness condition. Interestingly, bioactive treatment of material surfaces significantly affects their apatite formation. When all of the samples of cp-Ti and SLM-Ti were immersed in SBF, only the SLM-Ti MAH surface was fully covered by an apatite layer. The formation of bone-like apatite on the surface of SLM-Ti MAH is an indicator of a capacity for bonding with living bone when implanted in the body [34]. Apatite formation on Ti surfaces likely depends upon their surface charge rather than on the surface roughness and composition [18,42]. Kokubo et al. reported that cp-Ti with H_2_SO_4_/HCl and heat produced surface layers composed of rutile type titanium dioxide accompanied by a certain amount of the acid roots SO_4_^2−^ and Cl^−^. The treated metal formed apatite in SBF because of its positive surface charge that stimulates apatite nucleation by an accumulation of negatively charged phosphate ions and then positively charged calcium ions in the surrounding solution [18]. In this study, SLM-Ti MAH formed a similar surface layer composed of rutile accompanied by acid roots. Thus the mechanism of apatite formation on the SLM-Ti MAH in SBF may be understood as described above. In contrast, the SLM-Ti MA with titanium hydride surface layers did not form apatite in SBF because of its neutral surface charge [24,42].

The cellular responses were significantly affected by the surface roughness and wettability as well as surface charge [35,36,37]. The surface characteristics, apatite formation, and cell responses are summarized in Table 2.

In this study, the cells became strongly attached to the SLM-Ti surfaces via elongation and bridged the tops portion of the partially-melted Ti particles on SLM-Ti, and their elongation was further increased by MA or MAH treatment, even in as short a period as 0.5 h. This is in contrast with the performance of the cells of round morphology on cp-Ti. Furthermore, the cells became more branched and spread after 24 h on SLM-Ti MAH compared to other surfaces, as this cell morphology increased the contractility of the cytoskeleton and resulted in preferential osteoblast differentiation [43]. The differences in cell adhesion and morphology between cp-Ti and SLM-Ti are probably due to their surface roughness and hydrophilic properties. The capacity for cell attachment and spreading on roughened and hydrophilic surfaces is greater than on smooth and hydrophobic surfaces [44]. This is true in this case, in which SLM-Ti MA and SLM-Ti MAH were hydrophilic, with a contact angle ≤1° for both samples, while SLM-Ti U and cp-Ti were hydrophobic, with contact angles of 95° and 92°, respectively. In addition, cell adhesion is involved in stimulating signals that regulate cell proliferation, differentiation and migration [45]. In this study, the proliferation of MC3T3-E1 cells was enhanced as the surface roughness increased. On day 1, cell proliferation on the three SLM-Ti surfaces was shown to be significantly higher than that on cp-Ti, but there was no difference in cell proliferation among the three SLM-Ti surfaces. This result is consistent with a previous report by Tsukanaka et al. [21], in which the primary mouse osteoblast cell number was significantly increased on SLM-Ti compared to cp-Ti. The increased proliferation on SLM-Ti may come from an increase in initial adhesion of the preosteoblasts. These results suggest that the highly roughened topography of SLM-Ti provides greater support for the adhesion and proliferation of MC3T3-E1 cells. The tendency of increased proliferation on SLM-Ti groups was maintained for 3 days, although a significant difference emerged in the SLM-Ti groups. Proliferation in SLM-Ti MAH was comparable to SLM-Ti U, while proliferation in SLM-Ti MA was significantly lower than both SLM-Ti U and SLM-Ti MAH. A similar increase of proliferation, which was induced by heat treatment following acid treatment, was reported by Shi et al. [46]. In their study, Ti–6Al–4V alloy was subjected to heat treatment at 400 °C following mixed acid treatment of H_2_O_2_/HCl, and this increased the proliferation of MC3T3-E1 compared with the alloy subjected to the acid treatment alone. Because the surface roughness and hydrophilicity of SLM-Ti MA and SLM-Ti MAH were of the same degree, the greater cell proliferation on SLM-Ti MAH after 3 days may be attributed to rutile titanium dioxide having a positive surface charge.

The cell attachment and viability results indicate that the SLM-Ti MAH samples in the present study did not exert any cytotoxicity. The safety of bioactive treatment by mixed acid and heat had already been confirmed in an animal experiment in which the treated SLM-Ti mesh did not induce any inflammatory reactions after implantation in rats [47]. In addition, the treated SLM-Ti as well as the treated cp-Ti were shown to be able to tightly bind to living bone in vivo in a short period [4,28]. Although the bone bonding of the treated SLM-Ti is speculated to be attributable to its morphology being highly suitable for cell responses as well its enhanced capacity for apatite formation, the details of these cell responses have been unclear. Thus we investigated the effect of the surface roughness and mixed acid and heat treatment of SLM-Ti on the osteogenic differentiation of MC3T3-E1 cells. As shown in Figure 8, SLM-Ti MAH and SLM-Ti MA exhibited remarkably increased Alp after 10 and 14 days compared to SLM-Ti U as well as cp-Ti, indicating that the mixed acid and heat treatment stimulated osteoblast differentiation. In contrast, there was no difference in Alp activity between SLM-Ti U and cp-Ti at any of the tested time points. These results suggest that certain chemical factors, such as surface wettability, surface charge and apatite formation are of critical importance for the enhancement of Alp activity, rather than the surface roughness. In addition, we also found that calcium deposition in the extracellular matrix of the MC3T3-E1 cells was significantly increased in the SLM-Ti groups compared to cp-Ti, while SLM-Ti MAH had the highest calcium deposition. This indicates that the calcium deposition in the extracellular matrix is affected by both the surface roughness and chemical factors. The Alp produced by osteoblasts promotes matrix mineralization by increasing the concentration of phosphate ions and inhibiting phosphoric ester activity [48]. Enhanced osteogenic differentiation was also indicated by quantitative measurement of osteogenic-related gene expression by RT-PCR. As shown in Figure 10, the expression of Alp, Ocn, Opn and Runx2 was higher on SLM-Ti MAH than the three other surfaces. Runx2 protein is a critical osteogenic transcription factor activates the transcriptional activity of downstream osteogenic genes such as Alp, Ocn and Opn [49]. Expression of integrin-β1 was not different between the four surface types on day 7, but by day 14 it had decreased remarkably in SLM-Ti MAH. This may be explained by the shift from cell proliferation to differentiation that took place at this time. We also found that the expression of β-catenin and cyclin D1 was up-regulated by the SLM-Ti MAH surface. Beta-catenin is one of the proteins related to the Wnt signaling pathway [50] and this pathway specifically up-regulates the osteogenic regulator Runx2 [51]. Moreover, recent studies have reported that the signals from topographical cues activate the integrin-linked kinase (ILK)/β-catenin pathway to direct osteogenic cell differentiation [52,53]. In this study, the enhanced β-catenin and Runx2 activity suggests both Wnt and ILK signaling may play an important role in the osteogenic differentiation induced by SLM-Ti MAH. However, in the study by Shen et al. [54] using pathway-specific inhibitors it was found that osteogenic differentiation of preosteoblasts on Ti surfaces involved the MAPK/ERK signaling pathways. Osteogenic differentiation is a complex process involving numerous signaling pathways, so clarification of the mechanism for promoting cell differentiation on the bio-activated surfaces of SLM-Ti by mixed acid and heat is obviously needed.

Apart from the acid and heat treatment, alkaline and heat treatment is known to stimulate bone-bonding on Ti and already is in clinical use as total hip arthroplasty [26,27] and spinal fusion devices [55]. Ti treated with NaOH and heat has a lathlike surface layer composed of sodium titanate with nanometer roughness that exhibits hydrophilic characteristics and apatite formation in SBF [56]. Isaac et al. reported the osteogenic effect of NaOH-heat-treated Ti on fetal mouse osteoblastic cells isolated from calvaria [57]. RT-PCR showed up-regulation of Alp, Ocn and Runx2 on Ti receiving alkaline and heat treatment from days 3, 7, and 15, respectively. These up-regulation periods are comparable to those on SLM-Ti MAH in this study. On the other hand, the cell viability of the NaOH-heat treated Ti after 48 h was lower than that on untreated Ti, although it became the same after 7 days. This is in contrast with the increased cell proliferation on SLM-Ti MAH. This difference might be due to the difference in the surface roughness between the samples and also the sodium ion release from the NaOH-heat treated Ti [58]. The effect of sodium removal on the bone formation of Ti treated with NaOH and heat was reported by Takemoto et al., where the porous Ti treated with NaOH and heat displayed a lower osteoinductivity than the sodium-free Ti treated with NaOH, HCl and heat [17]. It was also shown by Yamamoto et al. that SLM-Ti mesh treated with the mixed and acid formed a greater amount of bone tissue than that treated with NaOH and heat [32].

In summary, a combination of SLM technique and bioactivation treatment with mixed acid and heat is a promising method for enhancing the biological activity of Ti owing to the high surface roughness, wettability, and apatite formation of the treated Ti. The bioactive SLM-Ti can be personalized fitted to the specific defect in each patient and should prove highly useful as a next generation dental and orthopedic implant.

## 5. Conclusions

This study has demonstrated that high surface roughness and additional bioactive treatment by a mixture of H_2_SO_4_/HCl and heat on SLM-Ti has a positive effect on enhancing preosteoblast differentiation. Bio-activated SLM-Ti has both a hydrophilic surface and apatite formation capacity, features which are critical for the bone-bonding of these materials. These features suggest this treated SLM-Ti surface has great potential as next generation orthopedic and dental implants.

## Figures and Tables

**Figure 1 nanomaterials-11-00987-f001:**
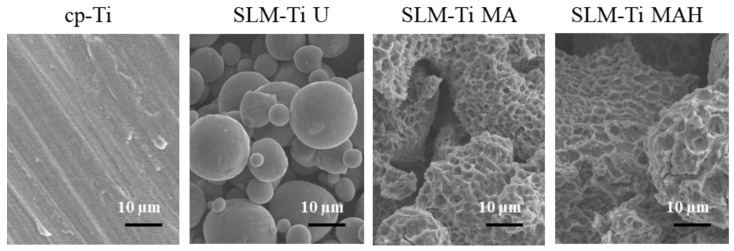
SEM images of cp-Ti and SLM-Ti before and after treatment with mixed acid and heat.

**Figure 2 nanomaterials-11-00987-f002:**
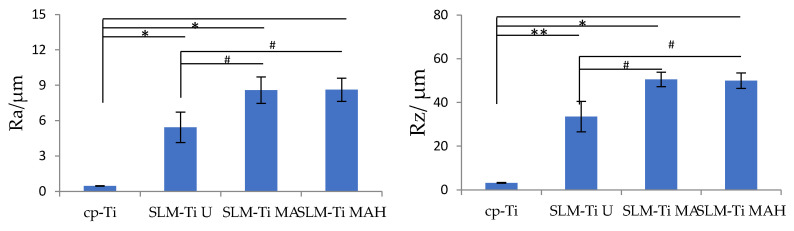
Surface roughness parameters of cp-Ti and SLM-Ti before and after treatment with mixed acid and heat. Ra: the average peak to valley distance, Rz: the highest peak to the lowest valley distance. Different from cp-Ti at * *p* < 0.05, ** *p* < 0.01; # different from SLM-Ti U at # *p* < 0.05.

**Figure 3 nanomaterials-11-00987-f003:**
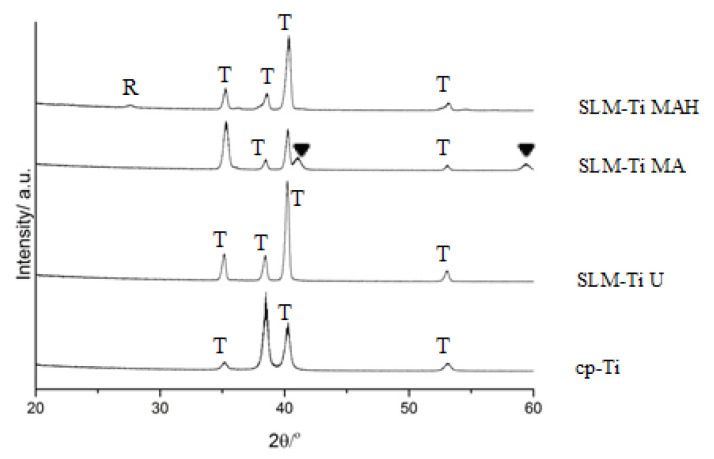
TF-XRD profiles of cp-Ti and SLM-Ti with or without the mixed acid and heat treatment. R: rutile, T: α-titanium, ▼: TiH_x_.

**Figure 4 nanomaterials-11-00987-f004:**
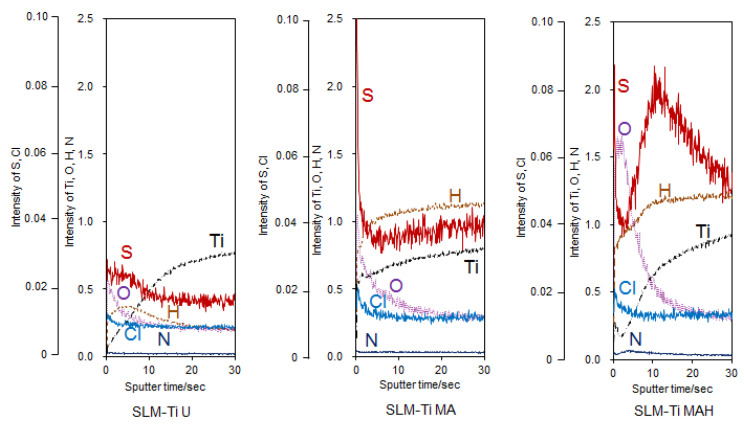
GD-OES profiles of SLM-Ti with or without the mixed acid and heat treatment.

**Figure 5 nanomaterials-11-00987-f005:**
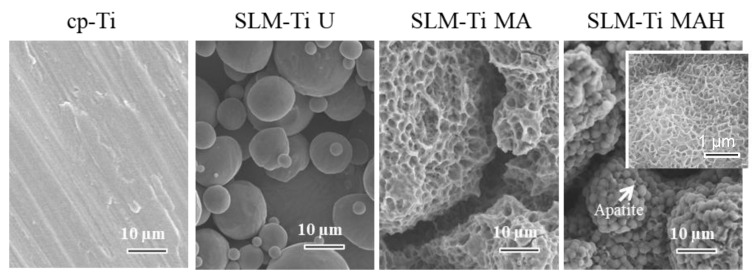
SEM images of cp-Ti and SLM-Ti after soaking in SBF for 1 day. Inserted window shows high magnification image. Only SLM-Ti MAH was subsequently covered by the apatite layer.

**Figure 6 nanomaterials-11-00987-f006:**
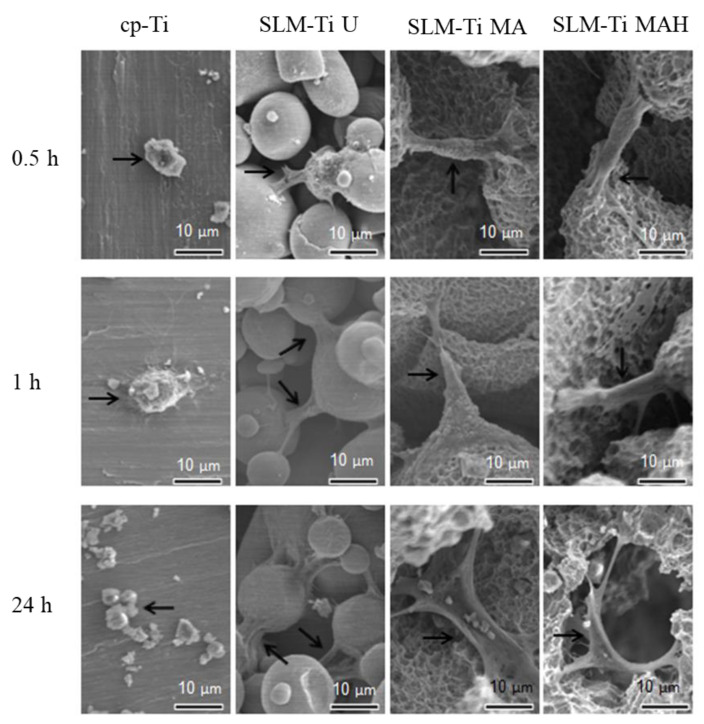
Morphology of MC3T3-E1 cells on cp-Ti and SLM-Ti surfaces after various incubation periods from 0.5 to 24 h. The black arrows indicate the cells.

**Figure 7 nanomaterials-11-00987-f007:**
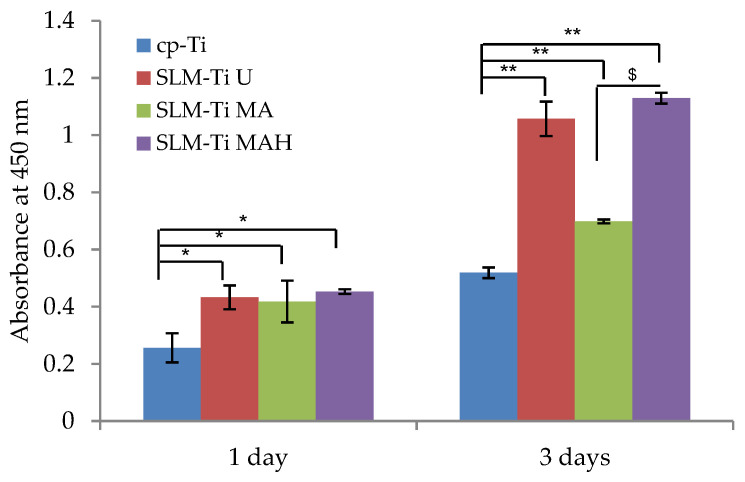
Viability of cells cultured on SLM-Ti that was untreated or treated with mixed acid and heat. Cells cultured on cp-Ti were tested as a control. After 1 and 3 days, the cell viability in each experiment or control sample was checked. All values are presented as the mean ± sd (n = 3), * different from cp-Ti, * *p* < 0.05; ** *p* < 0.01; $ different from SLM-Ti MA at *p* < 0.05.

**Figure 8 nanomaterials-11-00987-f008:**
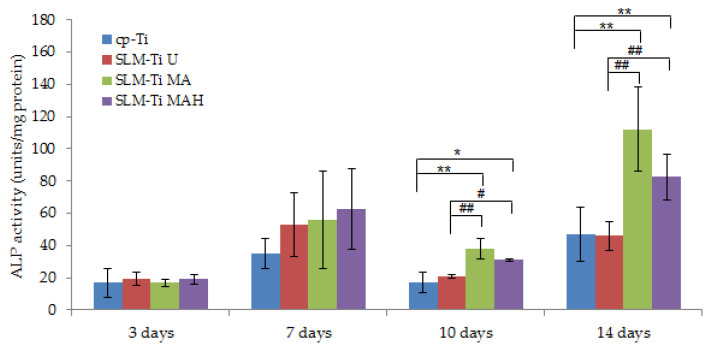
Alp activity of cells on SLM-Ti compared to the cp-Ti control after 3, 10, and 14 days. All values are presented as the mean ± sd (n = 6), * different from cp-Ti at * *p* < 0.05, ** *p* < 0.01; # different from SLM-Ti U at # *p* < 0.05, ## *p* < 0.01.

**Figure 9 nanomaterials-11-00987-f009:**
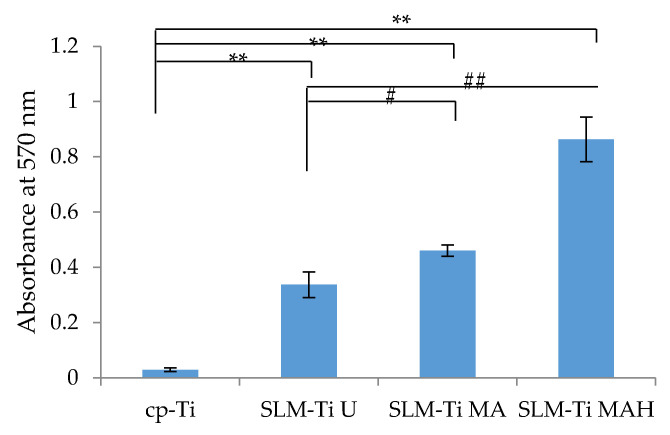
Mineralized matrix in MC3T3-E1 cells cultured for 21 days on the cp-Ti and SLM-Ti as measured by Alizarin red-S. (n = 3), ** different from the cp-Ti at *p* < 0.01, # different from SLM-Ti U at # *p* < 0.05, ## *p* < 0.01.

**Figure 10 nanomaterials-11-00987-f010:**
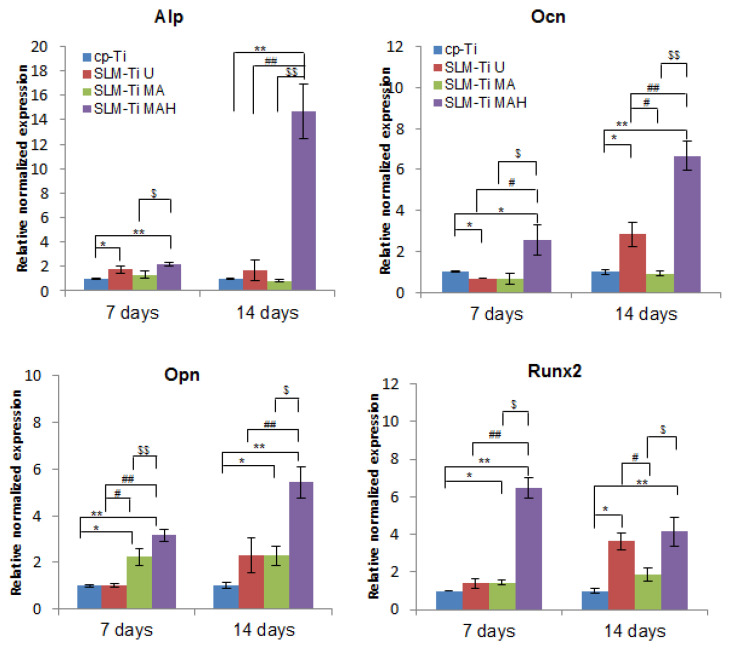
Real-time PCR for osteogenic gene expression of MC3T3-E1 cells cultured on cp-Ti and SLM-Ti with or without bioactive treatment at 7 and 14 days. The data were normalized to GADPH. All values are presented as the mean ± sd (n = 3), * different from cp-Ti at * *p* < 0.05, ** *p* < 0.01; # different from SLM-Ti U at # *p* < 0.05, ## *p* < 0.01, $ different from SLM-Ti MA at $ *p* < 0.05, $$ *p* < 0.01.

**Figure 11 nanomaterials-11-00987-f011:**
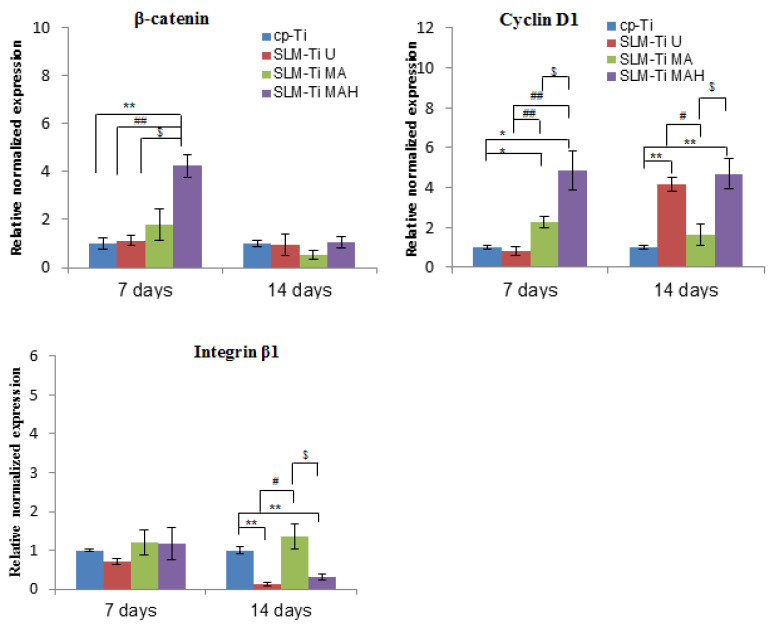
Real-time PCR for gene expression of MC3T3-E1 cells cultured on cp-Ti and SLM-Ti with or without bioactive treatment at 7 and 14 days. The data were normalized to GADPH. All values are presented as the mean ± sd (n = 3), * different from cp-Ti at * *p* < 0.05, ** *p* < 0.01; # different from SLM-Ti U at # *p* < 0.05, ## *p* < 0.01, $ different from SLM-Ti MA at *p* < 0.05.

**Table 1 nanomaterials-11-00987-t001:** Primer sequence of used genes.

Gene	Primer Sequence (F: Forward; R: Reverse; 5′–3′
β-catenin	F: GCCTACCACCAGCAGAATGT
	R: GAGGTGGCTGGGACTGTG
Integrin β1	F: TTGGGATGATGTCGGGAC
	R: AATGTTTCAGTGCAGAGCC
Cyclin D1	F: TTTCTTTCCAGAGTCATCAAGTGT
	R: TGACTCCAGAAGGGCTTCAA
Runx2	F: CCACAAGGACAGAGTCAGATTACA
	R: TGGCTCAGATAGGAGGGGTA
Alp	F: ACTCAGGGCAATGAGGTCAC
	R: CACCCGAGTGGTAGTCACAA
Ocn	F: AGACTCCGGCGCTACCTT
	R: CTCGTCACAAGCAGGGTTAAG
Opn	F: GGAGGAAACCAGCCAAGG
	R: TGCCAGAATCAGTCACTTTCAC
GADPH	F: TGTCCGTCGTGGATCTGAC
	R: CCTGCTTCACCACCTTCTTG

**Table 2 nanomaterials-11-00987-t002:** Surface characteristics, apatite formation, and cell responses of cp-Ti and SLM-Ti with or without mixed acid and heat treatment.

Sample	Roughness	Phase	Contact Angle	Apatite Formation	Cell Attachment	Cell Viability	Cell Differentiation
cp-Ti	smooth	α-Ti	95°	−	+	+	+
SLM-Ti U	rough	α-Ti	92°	−	++	++	+
SLM-Ti MA	rough	α-Ti +TiH_x_	<1°	−	++	+	++
SLM-Ti MAH	rough	α-Ti +rutile	<1°	+	+++	++	+++

## Data Availability

MC3T3-E1 cell line was obtained from ATCC, Manassas, VA, USA.

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
