# Peer review of "Bioactivation Treatment with Mixed Acid and Heat on Titanium Implants Fabricated by Selective Laser Melting Enhances Preosteoblast Cell Differentiation"

_nanomaterials, 2021, doi:10.3390/nano11040987_

Round 1

Reviewer 1 Report

The main objective of the work described in this paper is the enhancement of the biological properties of titanium scaffolds shaped by selective laser melting towards pre-osteoblastic cells. Mixed acid and heat treatment were used in order to enhance roughness, besides this treatments modified greatly some other surface properties such as wettability, titanium phases and surface element distribution. Ability of the material to support apatite formation after immersion in SBF was tested.

The impact of these modifications on murine pre-osteoblastic cell behavior was then assayed using a classical methodology. The discussion is well and carefully written making links between material surface changes and cellular response that bring important insight in the understanding of the cell-material interface.

I have some remarks concerning the methodology and the description of the results in order to strengthen the conclusions made by the authors:

  1. in the paragraph 2.4.”Cell culture”, the authors have written that the MC3T3-E1 cells were cultured in MEM (minimal essential medium). The usual culture medium for this cell line is the alpha-MEM without ascorbic acid that is an inducer of osteogenic differentiation. Can the authors check this part and, if the medium is different form Alpha MEM, justify this choice ?
  2. In order to guarantee the reproducibility of this work, the author should indicate the final cell density on the material surface: indeed, the cell concentration in each milliliter of culture medium is given but not the volume added by well in culture plate (12-well plate are generally filled by 1 to 2 mL of culture medium). I guess, when reading the other paragraphs including cell seeding that the wells contained 1mL of medium but for the readability, please homogenize the way to describe cell density at seeding.
  3. ALP assay was performed in an osteogenic medium whereas the evaluation of osteogenic gene expression was performed in growth medium that not induce differentiation. Methodologically, the choice of an osteogenic or a non osteogenic medium will give different indication. Indeed, the modification of differentiation behavior of cells in an osteogenic medium may reflect a passive facilitation of the process that is already induced elsewhere while in an osteogenic medium, the result may suggest that the tested material has an active impact on the differentiation process. I agree that this discrepancy, in the present work, should not greatly modify the discussion about the impact of surface properties of the different titanium scaffolds but the author should mention this methodological choice in the discussion and justify it. Moreover, as the assay for osteogenesis related genes was performed in a non-osteogenic medium, a positive control using cells with induced differentiation should be performed in order to apprehend the extent of the gene expression induction of the most effective material.
  4. In the paragraph 3.2 “Capacity for apatite formation”: the formation of a layer on the SLM-Ti MAH sample after immersion in SBF (figure 5) is visible by comparison with the figure 1. However, the clarity of this result may be improved by adding magnified images. The authors claim that the layer is made of apatite that should be the case after SBF immersion but an EDS pointing of this layer should conclusively prove this affirmation.
  5. Cell proliferation was assayed using a formazan-based test that directly measures the cell metabolic activity. With the results, cell viability and proliferation are extrapolated for the amount of metabolically active cells. So, after a short time culture such as 1 day as described in figure 7 and paragraph 3.4. “Cell proliferation”, a metabolic activity assay will more reflect cell attachment extent and viability than proliferation by itself. While the authors have discussed this correctly in the discussion, they should modify the text of the “Results” part and in the legend of the figure 7 consequently.

Author Response

Thank you very much for your valuable comments. We were answered all your question and made some changed according to your advice. Please check in the attached file.

Reviewer 2 Report

The author evaluated the effects on preosteoblast proliferation and differentiation of titanium metal with a high degree of roughness prepared by SLM (SLM-Ti) treatment with mixed acid and heat. This study provides insight into the importance of the role of high surface roughness and bioactive treatment on cell proliferation and differentiation.
Although this article is recommended to be published, there are several points to be solved as listed below. 

In Figure 7, only 1 or 3 days were analyzed. Why didn't you analyze 7, 10, and 14 days?
In Figure 8, only 3, 10, and 14 days were analyzed. Why didn't you analyze 1 and 7 days?
Viability of cells cultured on SLM-Ti that were untreated or treated.....--->.......that was untreated or treated.....
Why didn't you do gene expressions in 3 and 10 days in Fig. 10, 11? You were able to find out more accurate cell-attached proliferation by conducting experiments.

title: Bioactive treatment by mixed acid and heat on Titanium implants fabricated by selective laser melting enhances preosteoblast cell differentiation --->  Bioactive treatment with...... 
Line 2: modified so as to ---> modified to
Line 51: fused in layer by layer ---> fused layer by layer 
Line 63: are able to treat  ---> can treat
Line 69: capacity so as to exhibit ---> capacity to exhibit
Line 74: a few micro-meters ---> a few micrometers 
Line 76: an extended period of time ---> an extended period
Line 166: by means of LabAssayTM ALP ---> using LabAssayTM ALP
Line 365: significantly effects their apatite ---> significantly affects their apatite 
Line 372: accompanied by certain amount  ---> accompanied by a certain amount
Line 424: a short period of time ---> a short period
Line 453: pathway so as to direct ---> pathway to direct
Line 456: in study by Shen et al. ---> in the study by Shen et al.
Line 475: on bone formation of Ti ---> on the bone formation of Ti
Line 51: fused in layer by layer ---> fused layer by layer 
Line 63: are able to treat  ---> can treat
Line 69: capacity so as to exhibit ---> capacity to exhibit
Line 74: a few micro-meters ---> a few micrometers 

Author Response

Thank you very much for your valuable comments. We were answered all your question and made some changes in the manuscript according to your advice. Please check in the attached file here and highlighted part in revised manuscript. 

We hope that our manuscript will meet your request. 

Reviewer 3 Report

This is a mell-designed study, there are only several minor comments:

Line 86. Please, correct H2SO4.

Line 135. CO2.

Line 145. 105.

Line 176. 105.

Lines 193-196. The selection of target genes should be justified in the Introduction.

Lines 201-205. This section must be improved. First, Tukey's test is a posthoc test, which follows the ANOVA analysis and is used for the pairwise comparisons. Second, the Authors must check the normal distribution of the data, which is a prerequisite for the application of this statistic.

Figures 10-11. The meaning of different symbols showing the statistical differences should be mention in the legends.

Author Response

(The authors gave the same response as above.)
